# Voxel-based versus network-analysis of changes in brain states in patients with auditory verbal hallucinations using the Eriksen Flanker task

Lydia Brunvoll Sandøy [1,2*], Katarzyna Kazimierczak[3], Frank Riemer[4,5], Alexander R. Craven[2,6], Lars Ersland[2,6], Lin Lilleskare[7], Erik Johnsen[7,8], Kenneth Hugdahl[2,7,9], Renate Grüner[1]

**1** Department of Physics and Technology, University of Bergen, Bergen, Norway, **2** Department of Biological and Medical Psychology, University of Bergen, Bergen, Norway, **3** Institute of Computer Science, Czech Academy of Sciences, Prague, Czech Republic, **4** Mohn Medical Imaging and Visualization Centre, Department of Radiology, Haukeland, University Hospital, Bergen, Norway, **5** Neuro-SysMed, Department of Neurology, Haukeland University Hospital, Bergen, Norway, **6** Department of Clinical Engineering, Haukeland University Hospital, Bergen, Norway, **7** Division of Psychiatry, Haukeland University Hospital, Bergen, Norway, **8** Department of Clinical Medicine, University of Bergen, Bergen, Norway, **9** Department of Radiology, Haukeland University Hospital, Bergen, Norway

\* lydia.sandoy@uib.no

## Abstract

The present functional magnetic resonance imaging (fMRI) study investigated neural correlates of switching between task-processing and periods of rest in a conventional ON-OFF block-design in patients with auditory verbal hallucinations (AVHs) and healthy controls. It has been proposed that auditory hallucinations are a failure of top-down control of bottom-up perceptual processes which could be due to aberrant up- and down regulation of brain networks. A version of the Eriksen Flanker task was used to assess cognitive flexibility and conflict control. BOLD fMRI with alternating blocks of task engagement and rest was collected using a 3T MR scanner. The objective of the study was to explore how patients would dynamically modulate relevant brain networks in response to shifting environmental demands, while transitioning from a resting state to active task-processing.

Analysis of performance data found significant behavioral effects between the groups, where AVH patients performed the Flanker task significantly less accurately and with longer reaction times (RTs) than the healthy control group, indicating that AVH patients displayed reduced top-down guided conflict control. A network connectivity analysis of the fMRI data showed that both groups recruited similar networks related to task-present and task-absent conditions. However, the controls displayed increased network variability across task-present and task-absent conditions. This would indicate that the controls were better at switching between networks and conditions when demands changed from task-present to task-absent, with the consequence that they would perform the Flanker task better than the AVH patients.

**Data availability statement:** The data analyzed in this study is subject to the following licenses/restrictions: Due to restrictions posed by the ethics approval (REK Vest #2016/800), data are not allowed shared other than after request to the corresponding author, and after written permission from the Regional Committee for Medical Research Ethics in Western Norway (REK Vest). Requests to access these datasets should be directed to LBS, lydia.sandoy@uib.no or Christer Kleppe, data protection officer at Helse Bergen HF, Personvernombudet@helse-bergen.no.

**Funding:** 1) The European Research Council (ERC) grant #693124 (https://erc.europa.eu/homepage) to KH. 2) Helse-Vest grant #912045 (https://www.helse-vest.no/vart-oppdrag/vare-hovudoppgaver/forsking/forskingsmidlar/) to KH. 3) Trond Mohn Foundation grant #BFS2017TMT06 (https://mohnfoundation.no/en/) to RG. The funders had no role in study design, data collection and analysis, decision to publish, or preparation of the manuscript.

**Competing interests:** We have read the journal's policy and the authors of this manuscript have the following competing interests: KH, RG, ARC and LE own shares in the NordicNeuroLab Inc, https://www.nordicneurolab.com/, which produced add-on equipment for the fMRI data acquisitions. All authors declare no conflict of interest. This does not alter our adherence to PLOS ONE policies on sharing data and materials.

## Introduction

Auditory verbal hallucinations (AVH) are sensory experiences that occur in the absence of a corresponding external acoustic source [1]. It is a symptom that appears in patients with various diagnoses such as schizophrenia, bipolar disorder, depression, trauma-related disorders, dissociative disorders, personality disorders and Parkinson's disease [2,3]. AVHs are considered a hallmark symptom of schizophrenia and occur in about 70% of diagnosed schizophrenia patients [4,5]. Schizophrenia is a severe mental disorder that may be characterized by positive symptoms (e.g. hallucinations and delusions), negative symptoms (e.g. diminished emotional expression, social withdrawal and lack of motivation), and cognitive deficits such as deficits in attention and working memory [6]. Prominent symptoms of schizophrenia are AVHs which have been the subject of research interest in its own right since the early 20th century [7]. Different theories have been proposed for the underlying mechanisms of AVHs, such as an inner-speech model which attributes the voices to defective monitoring of inner speech [8], the traumatic memory model attributing AVH to deficits in inhibition and episodic/context memory failing to suppress recently activated memory traces [9,10], and aberrant bottom-up and top-down neuronal influences [11,12]. The self-monitoring theory explains AVH as deficits in internal self-monitoring mechanisms that compare expected with real sensations that arise from the patient's intentions [13]. Despite intense efforts the last decades in trying to understand the underlying mechanisms of AVH, we do still not have an explanation of why and how this phenomenon is more frequent in some patients and not others.

Numerous comprehensive studies have employed cognitive tasks, including dichotic listening [14], mental rotation [15], n-back tasks [16], Stroop [17], and the Eriksen Flanker task [18], in conjunction with functional magnetic resonance imaging (fMRI) to assess individuals experiencing AVHs. Schizophrenia patients show a larger Stroop effect than healthy controls [19]. It has been proposed that auditory hallucinations represent failure of top-down control of bottom-up perceptual processes [11]. This could be due to aberrant up- and down regulation of brain networks when environmental demands change from task processing to rest and vice versa, resembling fluctuations during the course of a normal day. The default mode network (DMN) is a collection of different brain regions that are commonly upregulated during passive rest with intrinsically driven modes of cognition [20]. This is in contrast to the extrinsic mode network (EMN), which is a generalized task-positive network that involves brain regions that are often shown to be upregulated during diverse cognitive tasks [21]. The dynamic up- and down-regulation of the DMN and EMN has previously been reported for both auditory [22], and visual tasks [23]. Patients with schizophrenia have been shown to have lower activation of a task-positive network, and increased activation in areas associated to the DMN [24,25]. Similarly, a study by Nygård et al. [26] reported that patients with schizophrenia failed to up-regulate task-positive networks and down-regulate the DMN during an auditory dichotic listening task. However, little is known about the interaction of these neural networks in patients experiencing AVHs. The Eriksen Flanker task [27] can be used to assess the ability to ignore task-irrelevant information, as participants are presented five arrows and have to indicate the direction of the central target (the middle arrow) while the surrounding arrows act as flankers [28]. A few studies have used the Flanker task on AVH patients together with fMRI. The study by Panagiotaropoulou et al. [29] found that increased reaction time (RT) variability was related to hypo-activation of the dorsolateral prefrontal cortex in schizophrenia patients. The study by Pappa et al. [30] compared patients with schizophrenia to patients with Parkinson's disease, where the former had reduced activation of the prefrontal and cingulate cortex indicating prefrontal hypofunction in patients with schizophrenia. Voegler et al. [18] found aberrant network connectivity during error processing between

networks associated to detection of errors and attention. Among the findings was aberrant connectivity between right anterior insula and inferior frontal gyrus and temporoparietal junction. A behavioral study by Smid et al. [31] used a letter version of the task in patients with psychosis with and without schizophrenia and found that the patients with psychosis performed the task similarly to healthy controls. To further investigate this relationship, we recruited patients with mild to severe degrees of AHVs to perform the Eriksen Flanker task while in the MR scanner, with the aim to assess cognitive flexibility and conflict control. The Flanker task is a classic task to investigate higher cognitive functions, with a large literature on its own. It has also been studied in schizophrenia and psychosis where it has been shown that patients with schizophrenia are impaired compared to healthy controls when solving the task. Using a visual task allowed us to go beyond the auditory modality inherent to AVH experiences, to capture higher cognitive functioning. By alternating blocks with task engagement (ON-blocks) and rest periods (OFF-blocks), we could investigate how patients would up- and down regulate the respective networks dependent on how environmental demands repeatedly changed from rest to active task-processing and back again. A voxel-based approach and network-analysis was used, as the former does not address interactions and network connectivity, while the network-analysis may have difficulty in examining the contribution of different nodes. An analogy for this is that the voxel-based approach allows us to identify individual steps, whereas the network-analysis reveals how these steps come together to form the overall structure of a staircase. By integrating these approaches, we can gain a comprehensive understanding of the dataset, effectively addressing the limitations inherent in each method.

We hypothesized that patients behaviorally would perform the Flanker task slower (RT) compared to controls and that task accuracy is similar to healthy controls. We assume that AVH is a failure of top-down control of bottom-up perceptual processes which could be due to aberrant up- and down regulation of brain networks. Thus, we hypothesized that patients and controls would differ significantly in active brain regions and use different networks during task- and rest periods.

## Methods

### Participants

The participants were 54 patients with mental disorders experiencing mild to severe auditory verbal hallucinations and 54 sex- and age-matched healthy controls (some of the data for the healthy controls were included in a previous publication by Craven et al. [32] for technical validation of functional MR spectroscopy findings). Participants were recruited from April 2017 until the 16th of December 2021. All participants provided written informed consent prior to participating in the study. They were free to withdraw from the study at all times with no consequences. Patients were recruited in the Bergen municipality area through the Vestland County Health Care System, primarily from the Sandviken Psychiatric Clinic, Haukeland University Hospital in Bergen, Norway. Health-care personnel were made aware of the project through announcements on boards, brochures, and social media. Some patients were recruited from psychiatric clinics in other counties in Norway. The project nurse contacted patients, organized transportation, conducted questionnaires and was present during the MR-scanning of the patients. Before inclusion in the study, patients underwent a PANSS (Positive and Negative Syndrome Scale) interview [33] no more than seven days before their MR scanning. The project nurse recruited patients who experienced auditory hallucinations. Only patients with a score of 3 or higher on the positive subscale score item 3 (P3 hallucinatory behavior) in the PANSS were recruited to the study. Furthermore, before scanning the project nurse administered the BAVQ (Beliefs About Voices Questionnaire, [34]) and the MVQ

(MiniVoiceQuestionnaire, [35]). These are self-report questionnaires about the emotional content of their voices (BAVQ), and questions related to daily AVH frequency and duration, environmental events preceding and following AVH episodes, the very first AVH episode, coping strategies, if the voice comes from the inside or outside of the head, if it is one's own voice heard, and whether the voice was present when filling out the questionnaire (MVQ).

Healthy control participants were recruited from the Bergen city area, at Haukeland University Hospital through posters and an article in a local newspaper (Bergens Tidende). Before inclusion, all participants were screened for previous history of major head injuries, medical implanted devices, substance abuse, neurological- and medical illnesses. Patient and control participants were compensated for their participation.

Out of the initially scanned participants, 4 patients were excluded due to an incomplete scanning session. This resulted in 50 patients, 20 women and 30 men, mean age 32.8 years (SD 9.7) with mean PANSS P3 score 4.6 (SD.8). The patients had different psychiatric diagnoses; n=29 with ICD-10 F20 (Paranoid schizophrenia), n=1 with F06.0 (Organic hallucinosis), n=4 with F12 and F19 (Drug-induced psychosis), n=1 with F23.4 (Acute paranoid psychosis), n=3 with F25 (Schizoaffective disorder), n=3 with F29 (Unspecified non-organic psychosis), n=1 with F31 (Bipolar disorder), n=1 with F32.3 (Severe depressive episode with psychotic symptoms), n=1 with F33 (Recurrent depressive disorder), n=4 with F60-F62 (Personality disorder), n=1 with unknown diagnosis, and n=1 with no diagnosis.

Healthy controls were matched to the patients according to sex and age (± 4 years), mean age 32.2 (SD 9.3) years. In the patient group, two patients were transgender, due to the unknown status of their transition, the sex they were born with was used for matching to a respective control participant. All participants were included in the voxel-based analysis (N=50), one patient and the corresponding control were excluded from the network-analysis due to an incomplete structural MR scan (N=49), and one patient was excluded due to no recorded responses for the behavioral analysis (N=49). The study was approved by the Regional Committee for Medical Research Ethics in Western Norway (REK Vest #2016/800). See Table 1 for further demographics.

### fMRI data acquisition

Participants were scanned using a GE 750 3T MR scanner (GE Healthcare), equipped with an 8-channel head coil. The initial structural $T_1$-weighted imaging used a Fast Spoiled Gradient Recall sequence (FSPGR), TE = 2.98 ms, TR = 6.8 ms, TI = 450 ms, acquiring 188 consecutive sagittal slices of 1 mm thickness, with 256 x 256 isotropic (1x1x1 mm) voxels. Subsequently, a block-design BOLD fMRI session was conducted, with in total 240 EPI volumes (TR = 2500 ms, TE = 30 ms, flip angle 90°, 1.72x1.72mm voxels, resolution 128 x 128, 36 slices of 3 mm thickness, total acquisition time 600s). Task-present blocks were 30 secs (12 volume scans) with trials presented every ~1500ms, and task-absent blocks were 60 sec (24 volume scans). The total time for the task-fMRI session was 10 minutes, with 120 Flanker stimuli presented

**Table 1. Demographic information about participants.**

|  |  | Sample size | Age | Sample male | Male age | Sample female | Female age |
|---|---|---|---|---|---|---|---|
| **Voxel-based analysis** | Controls | 50 | 32.2 ±9.3 | 30 | 33.3 ±9 | 20 | 30.6 ±9.8 |
|  | Patients | 50 | 32.8 ±9.7 | 30 | 34.3 ±9 | 20 | 30.5 ±10.6 |
| **Network- analysis** | Controls | 49 | 32 ±9.3 | 29 | 33 ±9 | 20 | 30.6 ±9.8 |
|  | Patients | 49 | 32.6 ±9.7 | 29 | 34 ±9 | 20 | 30.5 ±10.6 |

Age and sex of patients and controls. Mean ± standard deviation (SD).

throughout the 6 task-present blocks x 30sec plus 7 task-absent blocks x 60sec. Out of the total 120 flanker trials, 72 were congruent and 48 were incongruent trials (details below).

Finally, the MR session was completed with a traditional resting-state fMRI with a fixation cross of 8 min (TR = 2000 ms, 30 slices, other parameters were similar to the block-design BOLD fMRI run). This resting-state fMRI session was not used for analysis in the current paper.

## Eriksen Flanker task

A modified version of the Eriksen Flanker task was used. Five arrows were presented in the LCD goggles the participants were wearing (https://www.nordicneurolab.com/), where the middle arrow was the central target and the surrounding arrows acted as flankers. For each trial (ISI (interstimulus interval)= ~1500ms) the task for the participant was to indicate the direction of the middle arrow, using hand-held response grips (https://www.nordicneurolab.com/). For congruent trials, both the target and flanker arrows pointed in the same direction (>>>>> or <<<<<). For incongruent trials, the central target arrow pointed in a different direction than the surrounding flankers (>><>> or <<><<).

EPrime software version 2.0 SP1 (Psychology Software Tools Inc., Pittsburgh, PA, https://pstnet.com/) was used for visual presentation of the Eriksen Flanker task. During task-present blocks, the arrows were presented briefly and participants had to indicate the direction of the target arrow using response grips. A central fixation cross was seen in the goggles between stimuli and during task-absent blocks. The participants were instructed to respond as quickly as they could to indicate the direction of the middle arrow by pressing either the right or left index finger button on the response grips. Before the scanning, the participants went through a brief training run with instructions presented in the goggles. Participants did not receive feedback on their performance during the task.

Normality of data from the Flanker task was assessed using the Shapiro-Wilk test. For skewed distributions, Mann-Whitney U tests were used to compare patients and controls instead of independent sample t-tests. Two-way independent sample ANOVA was used to assess sex differences. A threshold of $p \leq 0.05$ was considered statistically significant in all comparisons.

## fMRI data preprocessing

The functional images were analyzed using SPM12 (UCL, London, UK, https://www.fil.ion.ucl.ac.uk/spm/) implemented in MATLAB 2020a (the MathWorks, Inc., Massachusetts, United States). The preprocessing steps for the EPI data included realignment, unwarping, normalization to the MNI template and smoothing. High-pass filtering was set to 190sec. The individual EPI-images from the first-level analysis were used to perform a SPM second-level analysis, where two paired t-tests were performed for the task-present and task-absent conditions, respectively, and with patients and controls as independent groups. If not specified otherwise, a threshold of $p< 0.05$ family wise error (FWE) corrected with a minimum of 10 voxels for cluster identification was considered statistically significant in all comparisons. A traditional SPM voxel-based region-analysis was complemented with a network connectivity analysis, using the CONN version 20b toolbox (https://web.conn-toolbox.org/, [36]). Preprocessing was conducted using the default CONN preprocessing pipeline which includes realignment and unwarping, outlier detection, segmentation and normalization to MNI-space and smoothing. For denoising a band-pass filter of 0.005Hz to 0.09Hz, realignment parameters (12), scrubbing (84), white matter, CSF confounds and main task effects for the task-present and task-absent condition was used. For the ROI-to-ROI functional connectivity

analysis, we used the default atlas implemented in CONN, which includes 132 brain regions, making up eight unique networks.

# Results

## Behavioral results

Shapiro-Wilk testing indicated that several variables related to task accuracy from the Flanker task had skewed distributions. Mann-Whitney U tests were therefore used to compare patients and controls instead of independent sample t-tests, these are summarized in Fig 1 and Table 2. There were significant differences between patients and controls, with large effect-sizes for overall response accuracy and congruent response accuracy, indicating that controls performed the Flanker task with higher accuracy compared to the patients. Furthermore, patients had significantly more missed responses compared to the controls (Table 2).

A factorial two-way ANOVA is considered robust to violations of normality, especially when the distributions are skewed in a similar manner. Therefore, this was used for analyses involving sex. The results are summarized in Table 3. The factor sex and the interaction of sex and group for correct left-hand incongruent responses were significant, $F(1, 94)=4.278$, $p =.041$. Analysis of simple main effects revealed that the male patients performed significantly better than the female patients ($p =.003$) and female controls performed significantly better than female patients ($p =.0004$).

## fMRI region-analysis results

**Task-present activations.** Paired t-tests contrasting task-present against task-absent periods showed statistically significant clusters exclusively for task-periods. Fig 2A-2B illustrates the significant task activations for patients and controls. Significant clusters for patients were located in the left (L) supramarginal gyrus, right (R) precentral gyrus, supplementary motor cortex (L), occipital fusiform gyrus (R), inferior occipital gyrus (R), bilateral middle frontal gyrus, anterior insula (L), bilateral thalamus proper, and in the ventral dorsal caudate (R). The healthy controls showed significant clusters in the bilateral supplementary motor cortex and middle frontal gyrus (L), for more details about clusters and locations, see tables in S1 Table and S2 Table.

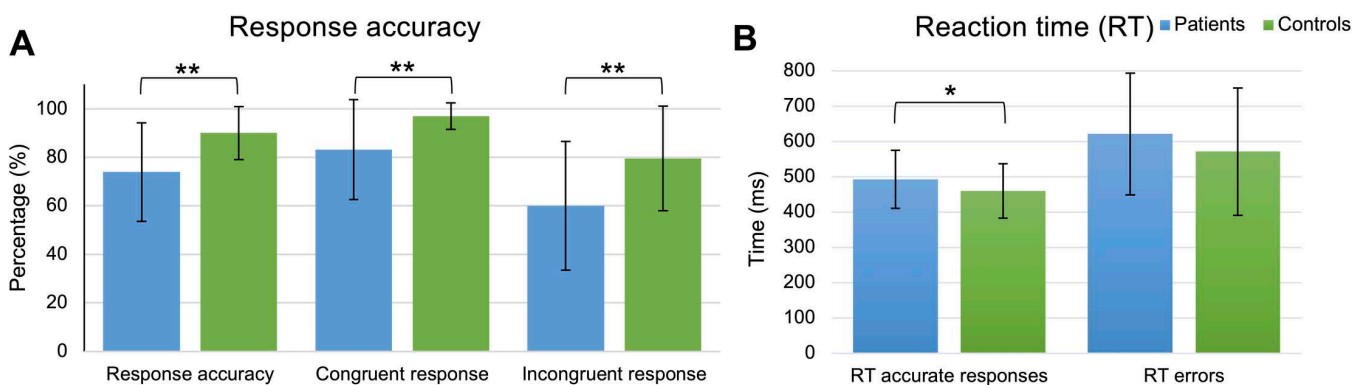

**Fig 1. Behavioral results from the Flanker task.** (A) Mean and standard deviation for overall response accuracy, congruent accuracy, and incongruent response accuracy. (B) Mean and standard deviation for reaction time (ms) for accurate responses and error responses for patients (blue) and controls (green). * $p <0.05$, ** $p <0.001$.

**Table 2. Summary of Mann-Whitney U tests comparing patients and controls.**

| Variables (percentage) | Controls | | Patients | | Z/ T | U | p | r/ η² |
|---|---|---|---|---|---|---|---|---|
| | M | SD | M | SD | | | | |
| Response accuracy | 90 | 10.9 | 73.9 | 20.3 | -4.9 | 520 | .000 | -0.5 |
| Missed responses | 3.2 | 3.9 | 8.1 | 11.6 | -2.3 | 453.5 | .021 | -0.27 |
| Congruent response accuracy | 97 | 5.5 | 83.2 | 20.6 | -5.5 | 452 | .000 | -0.55 |
| Incongruent response accuracy | 79.5 | 21.5 | 60 | 26.5 | -4.3 | 618 | .000 | -0.43 |
| Congruent - Incongruent | 17.5 | 18.8 | 23.2 | 22.8 | -2.3 | 891.5 | .020 | -0.24 |
| Accurate right hand responses | 90.9 | 10 | 75.2 | 20.9 | -4.9 | 528.5 | .000 | -0.49 |
| ⇒Congruent | 97.4 | 5.2 | 85.2 | 20.5 | -4.9 | 542 | .000 | -0.5 |
| ⇒Incongruent | 81 | 20.1 | 61.5 | 26.4 | -4.3 | 601.5 | .000 | -0.43 |
| Accurate left hand responses | 89.1 | 12.3 | 72.6 | 23 | -4.3 | 614 | .000 | -0.43 |
| ⇒Congruent | 96.6 | 6.2 | 81.1 | 24.2 | -4.7 | 571 | .000 | -0.47 |
| ⇒Incongruent | 78 | 24.8 | 60.9 | 28.5 | -3.5 | 712.5 | .000 | -0.35 |
| Mean RT accurate responses (ms) | 459.5 | 76.9 | 492.5 | 81.7 | -2.2 | 915 | .03 | -0.31 |
| Mean RT errors (ms) | 571.2 | 180.4 | 620.9 | 172.5 | -1.4* | | .164 | 0.02* |
| $\dfrac{\text{Response accuracy}}{\text{RT(ms)}}$ | 23.2 | 3.18 | 18.3 | 4.86 | 6* | | .000 | 0.27* |
| $\dfrac{\text{Errors}}{\text{RT(ms)}}$ | 2 | 2.88 | 4.5 | 4.58 | -3.8 | 687 | .000 | -0.53 |

M, mean; SD, Standard deviation; RT, Reaction time;

*Independent samples t-test was performed, T-value and eta squared was reported. Controls N= 50 and patients N = 49.

**Task-absent activations.** A paired t-test contrasting task-absent against task-present periods showed statistically significant clusters exclusively for task-absent periods, and Fig 2C-2D shows significant task-absent activations split for patients and controls. Patients showed significant clusters located in the bilateral precuneus, cuneus (R), bilateral middle occipital gyrus and angular gyrus (L). Controls showed significant clusters in anterior cingulate gyrus (L), medial frontal cortex (L), bilateral middle occipital gyrus, bilateral precuneus, cuneus (R), occipital pole (R), bilateral hippocampus, temporal pole (R), superior frontal gyrus (L), superior frontal gyrus medial segment (L) and fusiform gyrus (L). See tables in S3 Table and S4 Table for more details about clusters and locations.

Two-sample t-tests revealed no remaining significant clusters when patients and controls were contrasted against each other, for each condition separately. An additional region-analysis was conducted where congruent and incongruent trials were contrasted against each other. The analysis did not reveal any significant effects.

### Network-analysis.

Results from one-sample t-tests for the network connectivity analyses, based on CONN ROI-to-ROI analysis, are illustrated in Fig 3A-3D, with connectivity rings for task-present and task-absent conditions, respectively, for patients and controls separately. Each color line in the wheel represents significant t-values between the corresponding nodes, based on the average connectivity above (red) or below (blue) a given zero-level.

**Task-present condition.** One-sample t-tests were conducted separately for the patients and controls and are displayed as connectivity rings, where values in red indicate statistically significant positive t-values and values in blue indicate negative t-values between nodes of

**Table 3. Summary of two-way ANOVAs comparing patients and controls for sex-effects.**

| Variables (percentage) | Controls | | | | Patients | | | | F | p | Partial η2 |
|---|---|---|---|---|---|---|---|---|---|---|---|
| | Male | | Female | | Male | | Female | | | | |
| | M | SD | M | SD | M | SD | M | SD | | | |
| Response accuracy | 90 | 10.1 | 90.0 | 12.2 | 76.9 | 21 | 69.1 | 18.7 | 1.4 | .24 | 0.02 |
| Missed responses | 3.2 | 4 | 3.3 | 4 | 7 | 11.8 | 9.7 | 11.5 | 0.3 | .58 | 0.00 |
| Congruent response accuracy | 96.9 | 3.9 | 97.1 | 7.4 | 84 | 22.8 | 81.8 | 17 | 0.1 | .71 | 0.00 |
| Incongruent response accuracy | 79.5 | 21.3 | 79.5 | 22.4 | 66.3 | 25.1 | 50 | 26.2 | 2.8 | .1 | 0.03 |
| Congruent - Incongruent | 17.4 | 19.2 | 17.6 | 18.7 | 17.8 | 22.8 | 31.8 | 20.6 | 2.7 | .1 | 0.03 |
| Accurate right-hand responses | 90.9 | 9.4 | 90.8 | 11.2 | 77 | 23.5 | 72.4 | 16.2 | 0.5 | .5 | 0.01 |
| ⇒Congruent | 97.7 | 3.2 | 97.1 | 7.4 | 85.1 | 24 | 85.4 | 13.8 | 0.0 | .89 | 0.00 |
| ⇒Incongruent | 80.7 | 20.5 | 81.5 | 19.9 | 67.1 | 25.9 | 52.9 | 25.4 | 2.5 | .12 | 0.03 |
| Accurate left-hand responses | 89.1 | 11.6 | 89.3 | 13.5 | 76.8 | 21.5 | 65.8 | 24.2 | 2.3 | .14 | 0.02 |
| ⇒Congruent | 96.2 | 5.2 | 97.1 | 7.5 | 83 | 24 | 78.2 | 25 | 0.6 | .44 | 0.01 |
| ⇒Incongruent | 78.3 | 24.6 | 77.5 | 25.8 | 70 | 24.2 | 47.2 | 29.6 | 4.3 | .04 | 0.04 |
| Mean RT accurate responses | 471.2 | 46 | 441.9 | 107.2 | 505.4 | 76.7 | 472.3 | 87.3 | 0.0 | .91 | 0.00 |
| Mean RT errors | 574.3 | 148.1 | 566.5 | 224.4 | 651.9 | 163.6 | 571.8 | 179.1 | 1 | .32 | 0.01 |
| $\frac{\text{Response accuracy}}{\text{RT}}$ | 23.1 | 3 | 23.4 | 3.5 | 18.4 | 4.8 | 18 | 5.1 | 0.2 | .66 | 0.00 |
| $\frac{\text{Errors}}{\text{RT}}$ | 2 | 2.7 | 2 | 3.2 | 3.9 | 4.8 | 5.4 | 4.2 | 0.9 | .35 | 0.01 |

M, mean; SD, Standard deviation; RT, Reaction time; Partial $\eta^2$, partial eta squared. Controls N= 50 and patients N = 49.

the networks (Fig 3A-3B). Patients displayed strong intrinsic positive correlations between nodes of the salience network, F(2, 47)= 321.73, *p*<0.001. Statistically significant negative correlations were present between nodes in the salience network and the DMN (F(2, 47) = 19.05, *p*<0.001, especially for the anterior insula (R) and lateral parietal (L) nodes, T(48)= -7.53, *p*<0.001. Strong positive correlations between nodes of the dorsal attention and sensory-motor networks were also found.

Controls displayed a similar connectivity pattern as the patients. Strong intrinsic positive correlations between nodes of the sensory-motor network, F(1,48) = 494.29, *p*<0.001 and the salience network, F(2,47) = 318.06, *p*<0.001. Strong positive correlations between nodes of the fronto-parietal and the DMN networks, F(2,47) = 168.93, *p*<0.001, positive correlations between the fronto-parietal nodes of the lateral prefrontal cortex (LPFC) and posterior parietal cortex (PPC) and the lateral parietal (LP) node of the DMN. There was a negative correlation between the fronto-parietal node PPC (R) and DMN node in the medial prefrontal cortex (MPFC), and between nodes of the salience network and DMN.

**Task-absent condition.** Patients had strong intrinsic positive correlations between nodes of the salience network, F(2,47) = 797.61, *p*<0.001 (Fig 3C). Strong positive correlations between nodes of the dorsal attention and sensory-motor networks, F(2,47) = 177.86, *p*<0.001 were also found. There were negative correlations between nodes of the salience and the DMN networks. Controls displayed a similar connectivity pattern as the patients, with strong intrinsic positive correlations between nodes of the sensory-motor network, F(1,48) = 1337.47, *p*<0.001 (Fig 3D).

A two-sample t-test contrasting patients against controls directly, yielded significant negative correlations between nodes of the fronto-parietal network in the rest condition for the patients, F(3,94) = 7.25, *p* =.007.

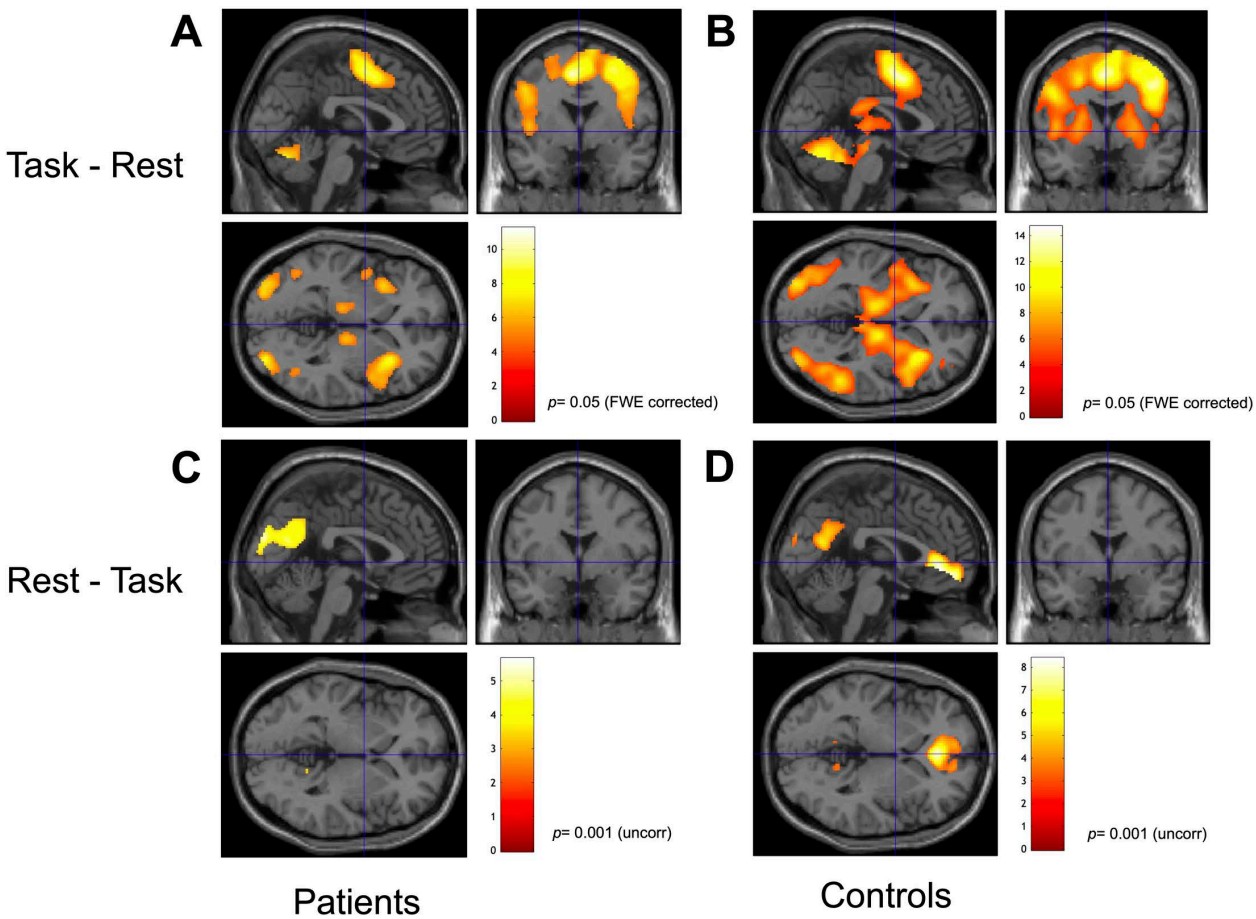

**Fig 2. Paired t-tests contrasting task-present and task-absent conditions against each other.** (A-B) Activations exclusive to task condition for patients and controls. (C-D) Activations exclusive to the rest condition for patients and controls. Cross-hair x, y, z, 0,0,0, *p* = 0.05 (FWE corrected), remaining figures *p* = 0.001 (uncorrected). Color bars shows t-values.

**Comparing the task-present and task-absent conditions directly.** A paired t-test contrasting the task-present condition against the task-absent condition showed that patients displayed an intrinsic negative correlation between the right lateral and left lateral nodes of the sensory-motor network, $F(2,47) = 22.12$, $p<0.001$, and positive correlations between nodes of the visual network and the sensory-motor network, $F(2,47) = 7.42$, $p_{uncorr} = 0.002$, as illustrated in Fig 4A. The controls displayed positive correlations between nodes of the DMN and cerebellar networks, $F(2,47) = 4.77$, $p =.047$ (Fig 4B). Positive correlations between nodes of the DMN and language networks $F(2,47) = 9.66$, $p =0.004$ were seen, except for a negative correlation between the posterior cingulate cortex and the right posterior superior temporal gyrus, $t(48) = -2.50$, $p_{uncorr} =.016$. The salience network was negatively correlated with the visual network, $F (2,47) = 6.61$, $p =.015$, and positively correlated with the language network, $F(2,47) = 6.89$, $p =.014$, as illustrated in Fig 4B.

## Discussion

The present study found significant behavioral effects associated with group affiliation, showing that patients performed the Flanker task significantly less accurately and had longer RTs than the healthy control group. In line with a previous study, the AVH patients had

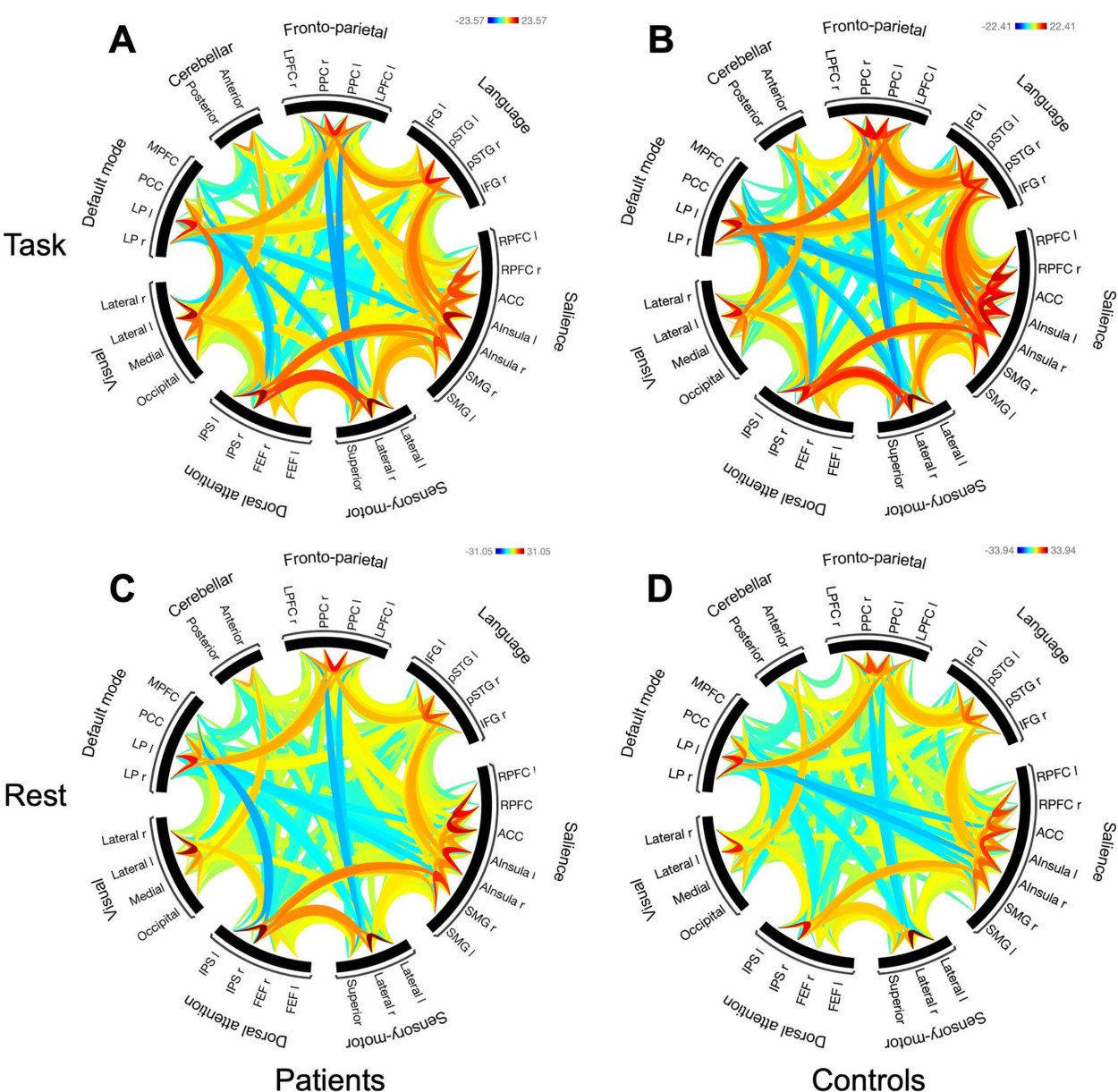

**Fig 3. One-sample t-tests from the network-analysis.** (A-B) Task activations for patients and controls. (C-D) Rest activations for patients and controls. Cluster threshold: *p* < 0.05 cluster-level p-FDR corrected and connection threshold: *p* < 0.05 p-uncorrected, two-sided. LP, Lateral parietal; PCC, Posterior cingulate cortex; MPFC, Medial prefrontal cortex; LPFC, Lateral prefrontal cortex; PPC, Posterior parietal cortex; IFG, Inferior frontal gyrus; pSTG, posterior superior temporal gyrus; RPFC, rostral prefrontal cortex; ACC, anterior cingulate cortex; AInsula, Anterior insula; SMG, supramarginal gyrus; FEF, frontal eye field; IPS, intraparietal sulcus; r, right; l, left.

significantly longer RTs during correct responses compared to the controls [37]. A meta-analysis of Flanker studies by Westerhausen et al. [28] provided strong evidence that psychosis patients displayed normality with regard to conflict control, and that schizophrenia patients' RT and ability to suppress competing responses were similar to controls [18,28,38]. However, our study found that AVH patients were significantly worse in conflict control. This could be due to our criteria for patient inclusion in the study, since only participants with mild to severe degrees of AVHs were included. Another reason for the discrepancy in our study, could

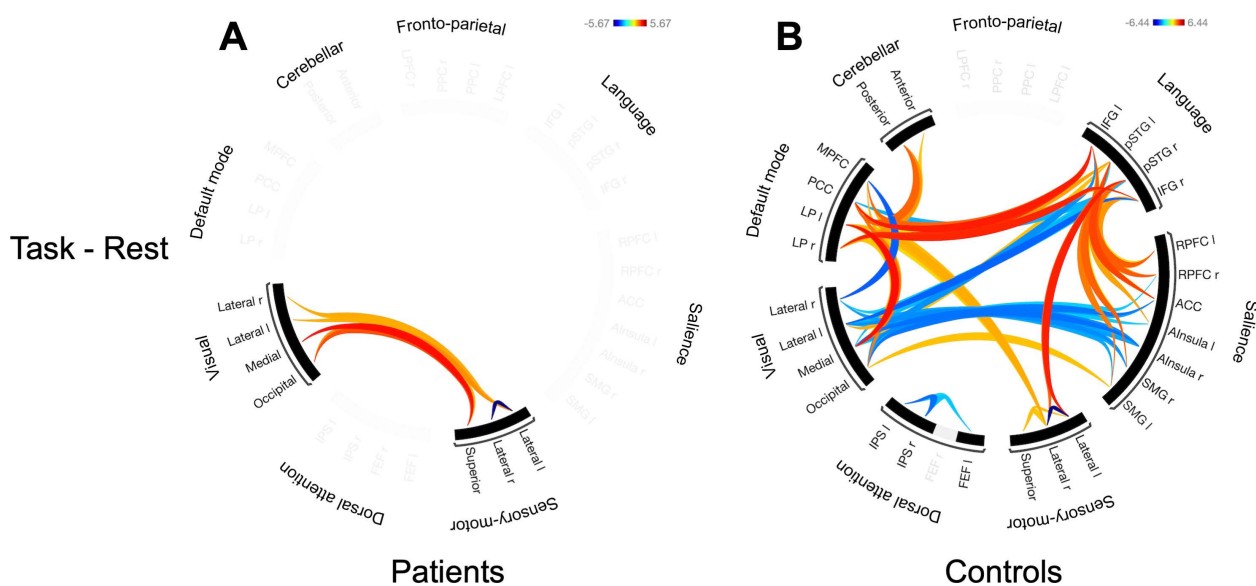

**Fig 4. Paired t-tests contrasting the task-present condition against the task-absent condition.** The connectivity rings shows significant correlations exclusively for the task-present condition. (A) Task activations for patients. (B) Task activations for controls. Cluster threshold: $p$ <0.05 cluster-level p-FDR corrected and connection threshold: $p$ <0.05 p-uncorrected. LP, Lateral parietal; PCC, Posterior cingulate cortex; MPFC, Medial prefrontal cortex; LPFC, Lateral prefrontal cortex; PPC, Posterior parietal cortex; IFG, Inferior frontal gyrus; pSTG, posterior superior temporal gyrus; RPFC, rostral prefrontal cortex; ACC, anterior cingulate cortex; AInsula, Anterior insula; SMG, supramarginal gyrus; FEF, frontal eye field; IPS, intraparietal sulcus; r, right; l, left.

be due to differences in the analyses performed. The meta-analysis by Westerhausen et al. [28] calculated the effect size of mean differences between patients and controls for incompatibility slowing, while the analysis in the current study focused on accuracy scores and RT related to accuracy.

The behavioral differences between the groups can be due to perceptual bottom-up mis-representations in the patient group. According to Hugdahl's [11] model, AVHs can be the result of perceptual bottom-up mis-representations. This can furthermore lead to a failure of top-down executive control which can result in a bias towards bottom-up processes and an exaggerated focus on the voices, which will interfere with the ability to tune in on other aspects of the environment. The top-down system may have problems prioritizing what information to process and what to put on hold. This is supported by the behavioral differences we observed between the patients and controls which were larger for incongruent trials. We suggest this is due to more cognitive efforts being required to inhibit the natural response of following the flankers in the incongruent trials.

The behavioral effects associated with group affiliation were not reflected in the BOLD voxel-based region and network connectivity analysis. This was somewhat unexpected, as we expected differences in neuronal activity between the groups. This could be caused by the fact that the cognitive task was a visual task. The network-analysis showed that both groups had similar activation of networks related to the task-present and task-absent conditions. However, comparing the conditions directly against each other revealed that the patient group activated these networks irrespective of whether the task was present or absent. This finding is in line with the study by Nygård et al. [26] in which patients with schizophrenia failed to up-regulate task-positive networks and down-regulate the DMN during dichotic listening task processing. In our study the controls showed more variability depending on whether the task was present or absent. There are two possible explanations for this finding. The first may be that AVH

patients show a general network hyper-activation and that they cannot switch between the networks when demands change. The second explanation may be that the patients show a general network hypo-activation and while they recruit all their resources for solving the task this is not very different from the activation seen during rest. These explanations can be reasons for the differences in performance and brain activation between patients and controls.

The voxel-based region analysis found clear activation patterns during the task-present condition that were consistent with activity of the EMN for both patients and controls. This is in line with previous studies, that the EMN is a task-positive network non-specific to the cognitive task being conducted [21,22]. A trend was found in the task-present condition with higher peak values in the control group compared to the patient group. This is somewhat similar to a meta-analysis that found lower activation in some task-positive network nodes in patients with schizophrenia across various tasks [25]. The DMN was active during the task-absent condition. Comparing congruent and the incongruent trials in the region analysis revealed no significant differences between the groups. This supports the notion of the EMN being a generalized task network, that the Flanker task regardless of the condition is challenging enough to elicit activation in brain areas associated to the EMN.

The main limitation of our study is the heterogeneity of the patient group. However, diversity and heterogeneity regarding diagnoses and possibilities for clinical misclassifications are always an issue with these complex diagnoses. One could argue that it would be easier to isolate AVHs in a diverse sample because other effects are canceled out. Thus, we chose to focus on a symptom that the patient group (mostly ICD-10 F20) had in common across the different sub-diagnoses, rather than a single diagnosis.

In our study, we used two different analyses to investigate how neural correlates of cognitive control during the Flanker task differ in patients with AVHs. However, future studies should investigate the dynamic up- and down-regulation of networks between task- and rest blocks, since such an approach could clarify whether patients show hyper- or hypo-activation of networks during changing environmental demands. Future studies could also explore additional visual tasks, to investigate if executive functions in the visual domain are globally affected in patients with AVHs, having the same patients conduct different tasks during fMRI, thus making it possible to compare task performance across tasks.

In conclusion, we found strong behavioral differences between the groups when performing the Eriksen Flanker task, where patients performed the task significantly worse than the healthy controls, indicating that AVH patients display reduced top-down guided conflict control. Due to this, the behavioral differences between the patients and controls were larger for incongruent trials as this requires more cognitive effort to inhibit the natural tendency of following the flankers and not the central target in the Eriksen task. The network-analysis showed that both groups had similar activation of networks related to task-present and task-absent conditions. However, the controls displayed increased network variability across task-present and task-absent conditions. This could mean that the controls were more flexible at switching between networks and conditions, with the consequence that they would also perform the Flanker task better than the AVH patients.

## Supporting information

**S1 Table. Task-present - task-absent activated clusters and peak voxel coordinates for patients. Summary of significantly activated clusters (with local maxima), and peak voxel x, y, z coordinates and corresponding t- and z-values and AAL atlas anatomical localizations, for the ON-OFF contrast.**
(DOCX)

**S2 Table. Task-present - task-absent activated clusters and peak voxel coordinates for controls.** Summary of significantly activated clusters (with local maxima), and peak voxel x, y, z coordinates and corresponding t- and z-values and AAL atlas anatomical localizations, for the ON-OFF contrast.
(DOCX)

**S3 Table. Task-absent - task-absent activated clusters and peak voxel coordinates for patients.** Summary of significantly activated clusters (with local maxima), and peak voxel x, y, z coordinates and corresponding t- and z-values and AAL atlas anatomical localizations, for the OFF-ON contrast.
(DOCX)

**S4 Table. Task-absent - task-present activated clusters and peak voxel coordinates for controls.** Summary of significantly activated clusters (with local maxima), and peak voxel x, y, z coordinates and corresponding t- and z-values and AAL atlas anatomical localizations, for the OFF-ON contrast.
(DOCX)

## Acknowledgments

We are grateful for the radiographers at Haukeland University Hospital: Roger Barndon, Christel Jansen, Turid Randa, Trond Øveraas, Eva Øksnes and Tor Erlend Fjørtoft, for their time and patience with data collection throughout this study. We also want to thank all of the patients and controls for their participation in the study.

## Author contributions

**Conceptualization:** Alexander R. Craven, Lars Ersland, Erik Johnsen, Kenneth Hugdahl.

**Data curation:** Alexander R. Craven.

**Formal analysis:** Lydia Brunvoll Sandøy, Katarzyna Kazimierczak, Frank Riemer.

**Funding acquisition:** Kenneth Hugdahl, Renate Grüner.

**Investigation:** Lydia Brunvoll Sandøy, Katarzyna Kazimierczak, Lin Lilleskare.

**Methodology:** Alexander R. Craven, Lars Ersland, Kenneth Hugdahl.

**Project administration:** Kenneth Hugdahl.

**Resources:** Kenneth Hugdahl.

**Supervision:** Kenneth Hugdahl.

**Visualization:** Lydia Brunvoll Sandøy.

**Writing – original draft:** Lydia Brunvoll Sandøy.

**Writing – review & editing:** Katarzyna Kazimierczak, Frank Riemer, Alexander R. Craven, Lars Ersland, Lin Lilleskare, Erik Johnsen, Kenneth Hugdahl, Renate Grüner.

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
