## [Decision Letter · Decision Letter 0]

10 Sep 2024

PONE-D-23-36278Voxel-based versus network-analysis of changes in brain states in patients with auditory verbal hallucinations using the Eriksen Flanker taskPLOS ONE

Dear Dr. Sandøy,

Thank you for submitting your manuscript to PLOS ONE. After careful consideration, we feel that it has merit but does not fully meet PLOS ONE’s publication criteria as it currently stands. Therefore, we invite you to submit a revised version of the manuscript that addresses the points raised during the review process. I apologyse for the long time needed to have a decision, but as you know it is difficult to find available and expert reviewers. Nevertheless, I reached two experts in this field and they kindly evaluated your work. As you can see, unfortunately, none of them is satisfied with the present version of your work, but they (and I) recognized the quality of the idea and the potential value of your manuscript. Therefore, I would ask you to consider their recommendations, including an explicit explanation of your sample (reviewer 1 is concerned about the comparison between patients and healthy controls, for example). 

We look forward to receiving your revised manuscript.

Kind regards,

Giulia Prete

Academic Editor

PLOS ONE

Journal Requirements:

“I have read the journal's policy and the authors of this manuscript have the following competing interests: KH, RG, ARC and LE own shares in the NordicNeuroLab Inc, https://www.nordicneurolab.com/, which produced add-on equipment for the fMRI data acquisitions. All authors declare no conflict of interest.”

4. We noted in your submission details that a portion of your manuscript may have been presented or published elsewhere. [The data for the healthy controls were previously used by our group in the study by Craven et al. (2023, doi:10.1002/nbm.5065).] Please clarify whether this [conference proceeding or publication] was peer-reviewed and formally published. If this work was previously peer-reviewed and published, in the cover letter please provide the reason that this work does not constitute dual publication and should be included in the current manuscript.

5. We note that you have indicated that there are restrictions to data sharing for this study. PLOS only allows data to be available upon request if there are legal or ethical restrictions on sharing data publicly. For more information on unacceptable data access restrictions, please see http://journals.plos.org/plosone/s/data-availability#loc-unacceptable-data-access-restrictions.

6. We notice that your supplementary [S1-S4 Table] are included in the manuscript file. Please remove them and upload them with the file type 'Supporting Information'. Please ensure that each Supporting Information file has a legend listed in the manuscript after the references list.

Reviewers' comments:

Reviewer's Responses to Questions

**Comments to the Author**

1. Is the manuscript technically sound, and do the data support the conclusions?

Reviewer #1: No

Reviewer #2: Yes

2. Has the statistical analysis been performed appropriately and rigorously? 

Reviewer #1: Yes

Reviewer #2: Yes

3. Have the authors made all data underlying the findings in their manuscript fully available?

Reviewer #1: No

Reviewer #2: No

4. Is the manuscript presented in an intelligible fashion and written in standard English?

Reviewer #1: Yes

Reviewer #2: Yes

5. Review Comments to the Author

Reviewer #1: The authors compared 50 patients with auditory hallucinations and 54 matched healthy controls using fMRI during performance of a cognitive task (the flanker task), which requires ignoring of irrelevant information. The patient group was heterogenous: while 32 had schizophrenia/schizoaffective disorder, the remainder had a variety of other diagnoses (organic hallucinosis, drug-induced psychosis, acute paranoid psychosis, unspecified non-organic psychosis, bipolar disorder, depressive disorder, personality and no or unknown diagnosis). Comparisons of task-related activations were carried out and also a functional connectivity analysis.

This study has two serious flaws. The first is the marked heterogeneity of the sample in terms of diagnosis. This represents an unusual strategy for studies in this field and severely limits the conclusions that can be drawn; inclusion of patients with organic psychosis, drug induced psychosis and unknown or no diagnoses are of particular concern here. The second is that comparing individuals with schizophrenia with auditory hallucinations and healthy controls is an inappropriate comparison - changes found could well be due to having schizophrenia rather than being related to presence of a particular symptom (in this case hallucinations). The appropriate comparison is between patients with a disorder who have/do not have a particular symptom. Clearly, this issue is complicated by the fact that the diagnosis was not schizophrenia in all the patients the authors included, but the underlying principle is the same.

Reviewer #2: In their manuscript “Voxel-based versus network-analysis of changes in brain states in patients with auditory verbal hallucinations using the Eriksen Flanker task“, Lydia Brunvoll Sandøy and co-authors set out to investigate neural correlates of switching between task-processing and periods of rest in a block-design in patients with auditory verbal hallucinations (AVHs) and healthy controls. They test cognitive flexibility and conflict control by use of a version of the Eriksen Flanker task in an fMRI block design alternating task engagement and rest.

Results show significant behavioural effects between patients and controls, with patients performing less accurately and with longer reaction times (RTs) than healthy controls. While both groups were shown to recruited similar networks during task and rest, controls displayed increased network variability across task-present and task-absent conditions, indicating that controls were better at switching between networks and conditions, thus explaining their better performance.

The topic of the study is timely and should be of interest to the broad readership of PLOS ONE. Furthermore, it has clinical relevance since, as the authors correctly state, the underlying mechanisms of AVH are still poorly understood.

Introduction

In general, the introduction is well written and provides a good overview over the topic. However, the description of the DMN, the EMN and their interaction in healthy participants should be described in more detail to give appropriate background. Is there general agreement on how DMN and EMN interact with each other during switching from task to rest periods? Are there any previous findings on DMN / EMN interaction in patients suffering from AVH?

Also, I’m missing a motivation as to why the Eriksen Flanker task was used here (as opposed to a variety of other tasks that have been used to study AVHs.

Given the title of the present manuscript the introduction should also provide some motivation as to why both voxel-based and network-analysis are employed here and advantages / disadvantages of both approaches should be eluted to.

Importantly, clear hypotheses based on the literature are missing at the end of introduction.

Minor points:

- What did Pappa et al. find when comparing patients with schizophrenia to patients with Parkinson’s disease and healthy controls?

Methods

Top of page 7: “All participants were included in the fMRI region-analysis, one patient and the corresponding control were excluded from the network-analysis due to an incomplete structural MR scan, and one patient were excluded due to no recorded response for the behavioural analysis.”

- Please correct typo: were / was

- If two patients were excluded, why are there still 50 patients in the voxel-based analysis group?

- Most importantly: Do the results look the same when the same 49 patients are included in the voxel-based and network analysis?

Minor points:

- “Only patients with a score of 3 or higher on the positive subscale score item 3 (P3 hallucinatory behavior) in the PANSS were recruited to the study.”: Did all these patients actually experience AUDITORY hallucinations?

- Why were different TRs used for the Task and RS session?

Results:

In the behavioural analysis, a sex difference in performance is identified, indicating that female patients’ performance is particularly low. It would be highly interesting to see whether this difference in performance is reflected in brain activation and connectivity patterns as well.

Minor points:

- For reader who are not familiar with the depiction of connectivity patterns as connectivity rings it might be helpful to describe in more detail what exactly these figures show.

Discussion:

Behavioural findings in the present study contradict previous findings from a meta-analytic study (Westerhausen et al.). It seems unlikely that this is only due to inclusion criteria. Could the authors please comment on further factors that might explain this discrepancy?

Altogether the discussion appears somewhat superficial. In particular, the voxel-based and network analyses should be discussed in conjunction rather than separately. Given the mostly unexpected brain imaging findings, further studies should be suggested that might shed more light on how neural correlates of cognitive control differ between patients and controls. Finally, a discussion of the limitations of the present study (which can also explain the null findings) is completely missing.

Altogether, this is a relevant study on a topic well suited for publication in PLOS ONE. Therefore, I would recommend publication of the manuscript, if my concerns can be adequately addressed by the authors.

6. PLOS authors have the option to publish the peer review history of their article (what does this mean? ). If published, this will include your full peer review and any attached files.

**Do you want your identity to be public for this peer review?** For information about this choice, including consent withdrawal, please see our Privacy Policy .

Reviewer #1: No

Reviewer #2: No

---

## [Author Response · Author response to Decision Letter 0]

21 Jan 2025

Response to reviewers are uploaded in the file "Response to Reviewers'.

---

## [Decision Letter · Decision Letter 1]

11 Feb 2025

Voxel-based versus network-analysis of changes in brain states in patients with auditory verbal hallucinations using the Eriksen Flanker task

PONE-D-23-36278R1

Dear Dr. Sandøy,

We’re pleased to inform you that your manuscript has been judged scientifically suitable for publication and will be formally accepted for publication once it meets all outstanding technical requirements.

Kind regards,

Giulia Prete

Academic Editor

PLOS ONE

Additional Editor Comments:

As you can see, one of the original Reviewer accepted to revise the manuscript and suggested its acceptance in the present form. The other Reviewer declined this new invitation, but I've reviewed the manuscript myself, and I believe the heterogeneity of sample is not an serious issue and, importantly, that the new paragraph in the Discussion is sufficient to solve this point. Thus, I am happy to accept the manuscript in the present form! 

Reviewers' comments:

Reviewer's Responses to Questions

**Comments to the Author**

1. If the authors have adequately addressed your comments raised in a previous round of review and you feel that this manuscript is now acceptable for publication, you may indicate that here to bypass the “Comments to the Author” section, enter your conflict of interest statement in the “Confidential to Editor” section, and submit your "Accept" recommendation.

Reviewer #2: All comments have been addressed

2. Is the manuscript technically sound, and do the data support the conclusions?

Reviewer #2: Yes

3. Has the statistical analysis been performed appropriately and rigorously? 

Reviewer #2: Yes

4. Have the authors made all data underlying the findings in their manuscript fully available?

Reviewer #2: Yes

5. Is the manuscript presented in an intelligible fashion and written in standard English?

Reviewer #2: Yes

6. Review Comments to the Author

Reviewer #2: The authors have addressed all my previous concerns. Therefore I consider the manuscript suitable for publication in it's current form.

7. PLOS authors have the option to publish the peer review history of their article (what does this mean? ). If published, this will include your full peer review and any attached files.

**Do you want your identity to be public for this peer review?** For information about this choice, including consent withdrawal, please see our Privacy Policy .

Reviewer #2: No

---

## [Editor Report · Acceptance letter]

PONE-D-23-36278R1

PLOS ONE

Dear Dr. Sandøy,

I'm pleased to inform you that your manuscript has been deemed suitable for publication in PLOS ONE. Congratulations! Your manuscript is now being handed over to our production team.

Kind regards,

on behalf of

Dr. Giulia Prete

Academic Editor

PLOS ONE